# Towards women's digital health equity: A qualitative inquiry into attitude and adoption of reproductive mHealth services in Bangladesh

**M. Jonayed** *, **Maruf Hasan Rumi**

Department of Public Administration, University of Dhaka, Dhaka, Bangladesh

* jonayedsj03@gmail.com

**Data Availability Statement:** The anonymous versions of the interview transcripts are provided in Supporting Information (S1 File).

## Abstract

Health equity in Bangladesh faces a large chasm over the economic conditions, socio-cultural factors and geographic location despite the push for digitalization of the health sector. While some research has been conducted assessing the viability of digital health solutions in Bangladesh, gender dynamics of digital healthcare have been absent. This study dived into healthcare equity for women with a focus on reproductive health services delivered through mobile devices. This paper reported the findings of a qualitative study employing in-depth interviews conducted among 26 women about their behavioral intention to use mHealth services for reproductive health and the underlying factors influencing this intention with the help of the Integrative Model of Planned Behavior (IMPB). A snowball sampling technique were used to interview those university educated women, aged 21–31, based on their familiarity and exposure of mHealth services from seven universities in Bangladesh. The findings suggested that users of mHealth services find it more convenient and secure compared to visiting healthcare facilities, especially for trivial issues and inquiries regarding their reproductive health. Although promoting such services is lagging behind traditional healthcare, the attitude toward reproductive health services in Bangladesh is generally favorable resulting increasing adoption and use. Because such information-related mobile services (apps, websites, and social media) served as a first base of knowledge on reproductive health among many young girls and women in Bangladesh, who are generally shy to share or talk about their menstruation or personal health problems with family members, peers, or even health professionals due to socio-cultural factors and stigmatization. Conversely, urban centric services, availability of experts, quality management, security of privacy, authenticity of the information, digital divide, lack of campaign initiatives, lack of equipment and technology, lack of sex education, and outdated apps and websites were identified as obstacles that constrain the widespread use of reproductive mHealth services in Bangladesh. This study also concluded that promotion will be crucial in reforming conservative norms, taboos, and misconceptions about women's health and recommended such endeavors to be initiated by the policy makers as there is a substantive need for a specific policy regulating emerging digital health market in Bangladesh. Notwithstanding, women-only sample, low sample size, narrow focus on mHealth users and absence of perspectives

**Funding:** The author(s) received no specific funding for this work.

**Competing interests:** The authors declare that no competing interests exist.

from healthcare providers were among shortcomings of this study which could be addressed in future research. Further quantitative explorations are must to determine the usage patterns of reproductive mHealth services and their effectiveness that would identify implementation challenges in terms of customization and personalization in reproductive healthcare in a developing country like Bangladesh.

## Author summary

In accordance with the vision of digitalized and universal healthcare, the government has attempted to incorporate various innovations and technology into the health sector in Bangladesh to promote accessibility and equity. However, implementation policies related to digital health received little attention and the adoption of such technologies have remained low, especially among women. The existing patriarchal social structure and stigmatization of women's reproductive health further discourage women from accessing physical healthcare infrastructures and services. Digital health, specifically mobile health, solutions are seen as a viable path toward achieving health equity in Bangladesh. Our paper explores the attitude of women toward mobile health services to understand the trend of mobile health uses and the challenges ahead for both policymakers and developers. We find the general environment to be favorable for the popularization of digital health solutions. For this, the role of the government is critical in setting out specific policies related to digital health in Bangladesh and similar contexts.

## Introduction

Health equity refers to individual agency or the ability to make choices and equitable access to required resources and opportunities for realizing one's well-being regarding physiological, emotional, and social health [1]. It is characterized by the absence of systematic healthcare system inequalities for citizens from different social strata [2]. In the age of digital health, it outlines the fair and just availability of digital healthcare, its outcomes, and equity in its solutions [3]. Nonetheless, the determinants of digital health equity are intricately linked to technological literacy and access and operate at the individual, interpersonal, community, and societal levels. From a gender perspective, promoting digital health without proper analysis in the design, evaluation, and implementation process of policies and programs regarding social norms and values can make inequity worse than before [4].

 Conversely, reproductive health constitutes an intrinsic component of the physiological well-being of women, yet it is frequently underestimated in developing countries because of socio-cultural values and norms [5]. Reproductive health care services encompass a range of essential services, such as maternal and newborn care, contraceptive services, prevention and control of sexually transmitted infections (STIs), prevention and treatment of infertility and cervical cancer, safe abortion care, comprehensive sexuality education, counseling, and care for sexual health and well-being [6]. Those services can either be delivered by physical interventions or using digital technology. But in a resource-poor setting, as well as amid the frequent emergence of infectious disease outbreaks, women's reproductive health and rights are at constant risk as a result of a reversal of priorities and availability, accessibility, and affordability of such reproductive services [7]. As a result, reproductive health is a burning issue for achieving health equity for women at the era of flourishing digital healthcare solutions.

Health equity in Bangladesh faces a large chasm over the socio-economic condition and location. Both rich and urban residents get better services, while the poor population faces multiple barriers related to geography, social conditions, and accessibility to healthcare services [8]. On the contrary, through community engagement, recent initiatives, specifically the operationalization of community clinics at the grassroots level, have enabled poor and marginalized groups, including women, to have greater access to health services. It ensures better health outcomes and equity, efficacy, safety, and timeliness at a much lower cost [9]. Besides, over 40 digital health initiatives include a wide range of activities about maternal health, drug abuse, HIV/AIDS, and general healthcare available in Bangladesh [10–13].

Notwithstanding digitalization endeavors, there is a lack of comprehensive evidence regarding the influence of technology on ensuring fair and equal access to healthcare services and information in the underprivileged areas of Bangladesh [12]. It has been recorded that Bangladeshi women are more vulnerable to STIs because of their limited awareness, and they have a greater likelihood of experiencing unintended pregnancy in comparison to neighboring countries [14,15]. Culture and religion, through belief systems, values, norms, psychology, and superstition, play a dominant role in women's knowledge and decisions on reproductive health, especially in rural areas of Bangladesh [16]. In addition to sociocultural predispositions, women may face barriers, including limited healthcare infrastructure and services, inadequate medical personnel, and unsatisfactory educational and promotional programs [17]. The Bangladesh Demographic and Health Survey from 2017–2018 revealed that a significant majority (67%) of women between the ages of 15 and 49 encountered challenges when attempting to obtain healthcare services [18]. Furthermore, stigmatization of reproductive health including menstruation, contraceptive use, abortion, sex education and counselling dissuade women from seeking health services despite advancements in economic and political empowerment of women in Bangladesh.

Analysis of data from 46 low- and middle-income countries (LMICs), including Bangladesh, reveals that there has been a notable enhancement in the fairness of reproductive health services. This improvement is evident in various indicators, such as family planning, fertility, antenatal healthcare, infant mortality, and reduced proximity to urgent medical assistance [19]. It was concluded from the data that mobile outreach could further close the equity gap between the rural and poorer urban residents. Another study conducted in Bangladesh found notable disparities in providing antenatal care by trained professionals and access to delivery facilities based on wealth-related factors [2]. Notwithstanding, in a digital health service delivery context, a strong association with gender disparity was discovered because women are less likely to use digital services in Bangladesh [20]. To ensure universal health coverage, attention to the digital divide is essential [21]. Although the research into digital health, including mHealth in developing countries like Bangladesh, is growing in numbers, there are minimal studies taken into account the issues of digital health equity for women [4,22,23]. Also, evidence of mHealth utilization is scarce for accessing health services in Bangladesh [24]. Therefore, it is crucial to comprehend women's attitudes and adoption behaviors concerning digital health, mainly related to reproductive mHealth solutions, before implementing customized services for them and enhancing the equity situation. This paper presents the results of a qualitative study among young adult women regarding their intention to use mHealth services for reproductive health and the underlying factors influencing this intention. Through the exploration of their attitude, it would be easier to assess whether such services could be viable alternatives to current reproductive health practices among young girls and women in Bangladesh. The end objective is to aid policymakers, professionals, and influential stakeholder groups in the health sector to enhance women's fair and just access to reproductive health care and promote health equity.

## mHealth as a vehicle to ensure equity for reproductive health in Bangladesh

The proliferation of mobile devices, including smartphones and wearables, in conjunction with the technological revolution in health communications and the progress of health sciences, has propelled the notion of mobile health (mHealth) to prominence within digital health solutions. The platform allows users and healthcare professionals to conveniently retrieve, track, and oversee data while also serving as a means for promoting health, responding to emergencies, providing support at the point of care, and gathering data [25,26]. Also, it offers a superior influence on face-to-face interventions at a low cost [27] and has the potential to surmount obstacles in developing countries and transform global healthcare service delivery to achieve sustainable development goals [28,29]. In the same context, the National Digital Health Strategy (NDHS) 2023–2028 is being undertaken in Bangladesh [30], with a strategic objective to make services available on digital platforms and through mobile apps [31]. The *Shasto Batayon 16263* (National Health Helpline of Bangladesh, operated around the clock to provide general health information and health-related emergency services), which incorporates sophisticated interactive voice response (IVR) technology into a unified national health call center, is one of the most successful mHealth services in Bangladesh, connecting citizens to every district and Upazila-level government hospitals [32,33]. Other mHealth services are also becoming successful in improving knowledge and awareness, such as intervention for Diabetes [34–36], Hypertension [37], Post-menstruation regulation contraceptive use [38], and Pregnancy surveillance [39].

Nevertheless, with a dramatic growth of mobile phones in the last decade thanks to digitalization efforts, Bangladesh now has more than 100 percent subscriber concentration [40], and smartphone adoption is booming rapidly [41]. This development offers a promising prospect for the implementation of smartphone-based mHealth services in Bangladesh, where it is perceived as a catalyst for progress in catering to underserved markets [42] and can bridge the personnel and distance gap in support of reproductive health [43]. While mobile phone ownership and access are comparatively lower among women than men in low and middle-income countries (LMICs) [44], understanding the gender dynamics and power relations in digital health programs, including mHealth, can convalesce health outcomes. Besides, increasing the availability of reproductive health services and ensuring their utilization is not only essential for the overall well-being of women as part of universal access to healthcare but also necessary for lowering mortality during births and for preventing and controlling STIs in developing countries, including Bangladesh. Despite a growing demand for easily accessible reproductive healthcare systems, there are apprehensions that the outcome of mHealth services is not clearly understood [45]. The analysis of another study also showed an unsatisfactory condition of the digital health service sector and the mHealth sector's usage is very small compared to the entire healthcare in Bangladesh [46]. To understand this phenomenon, there is a necessity to explore the attitude of people especially women about their perception and adoption of reproductive mobile health services. Also, existing research primarily focus on the general health service sector in Bangladesh and do not focus on the reproductive health interventions for young girls and women. The socio-cultural context in Bangladesh creates impediments for young women, especially unmarried one to seek out reproductive care and mHealth can remove such hurdles to create a healthcare system free from any inequality.

## Theoretical framework of the study

Multiple theories have been developed to elucidate and forecast various behaviors within health communication research because each behavior is unique from some perspectives.

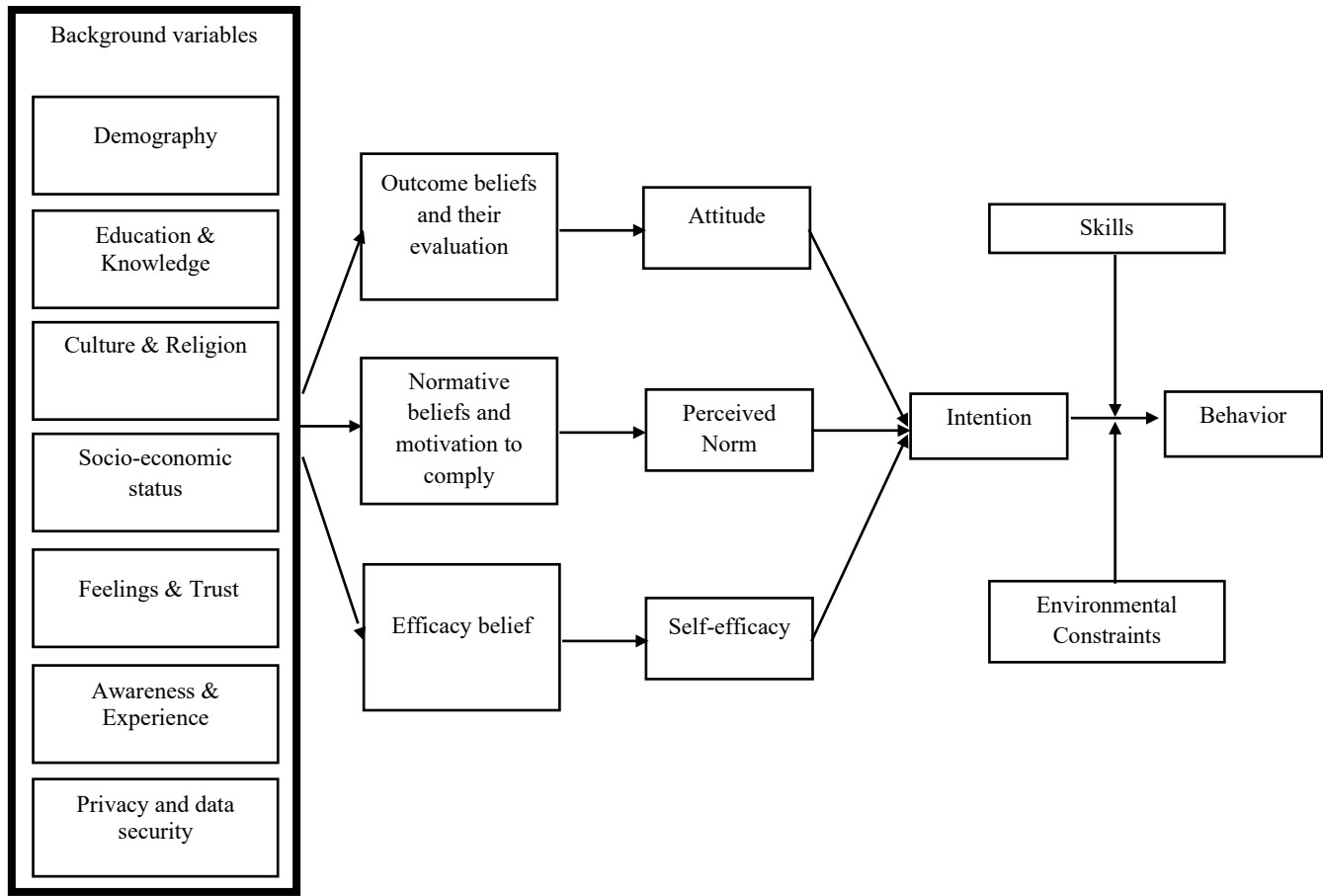

**Fig 1. An Integrative Model of Planned Behavior (Adapted from Yzer [51]).**

Stemming from the theoretical roots of the Theory of Reasoned Action (TRA) [47] and the Theory of Planned Behavior (TPB) [48], The Integrative Model of Behavioral Prediction (IMBP) has been widely used and tested in the domain of health among many other fields [49]. This study used the IMBP model because it incorporates elements from both the TRA and the TPB to create a comprehensive framework that includes additional details and variables, enabling a more accurate prediction and comprehension of human behavior. It stipulates that although numerous variables could influence behavior, only a tiny proportion is necessary to understand the performance of intended behaviors in a given population [47,50]. Specifically, the intention to perform certain behaviors comes from reasonable beliefs about that behavior, but not always rationally. Three constructs were hypothesized to influence the intention to behave: attitude, perceived norms, and self-efficacy. Together with skills and environmental constraints, they influence behavioral performance. Attitude is the general sense of the outcome being good or bad, the perceived norm is the expected societal pressure which can be supported by social networks (injunctive) and extent of conformation (descriptive), and self-efficacy is the presumed competence of oneself to execute the behavior [51]. In an intention-behavior relationship, the IMPB (see Fig 1) posits that individuals are more likely to act on their intentions when they possess the requisite skills and when environmental factors do not hinder their ability to perform the behavior [52].

Incorporating background variables in the IMPB enables elasticity and malleability according to different cultures and contexts. It also recognizes that beliefs that shape behavior are grounded in demographic, socio-economic, and cultural factors. While those can be possible belief sources, they do not necessitate theoretical associations with the constructs [51]. Nevertheless, this conceptualization could help us explain the behavior based on participants' attitudes and other variables that influence behavioral intention in the background of acceptance and adoption of mHealth for reproductive services against the backdrop of health equity.

## Materials and methods

This research employed a qualitative approach to investigate the behavioral intention of women towards reproductive mHealth services and understand the underlying factors of adoption and acceptance of these services. Qualitative research is deemed appropriate for a topic that is not well understood and sensitive, such as this study [53]. Also, in the field of health communication, qualitative research has gained significant acceptance and prominence among scholars [54]. Qualitative inquiries in health communication research are formulated utilizing various theoretical frameworks to enhance understanding of the difficulties encountered in the provision of healthcare and health promotion [55]. The qualitative research approach offers maximum interaction [56] that helps researchers understand the participants' attitudes and behavioral intentions. Similar methods have been adopted for mHealth research in various contexts [57,58].

### Study context

This study was conducted in the months of April—May 2023 in seven public universities across Bangladesh, namely the University of Dhaka, Bangladesh University of Professionals, Jahangirnagar University, Jagannath University, National University, Begum Rokeya University, and Bangabandhu Sheikh Mujibur Rahman Science & Technology University. The first four universities are inside Dhaka, while the other three are in three different districts of Gazipur, Rangpur, and Gopalganj [59]. Researchers selected youths (aged 18–35 according to National Youth Policy of Bangladesh [60]) with at least an undergraduate educational background as they are more likely to be exposed to mHealth services because of understanding of their body and knowledge of mobile technologies. Moreover, choosing various locations to collect data was meant to negate any urban biases (for example residents in Dhaka city are perceived to be more technologically advanced but have less conservative social values than its peripheral and rural areas) and produce a representative study. Thus, such diversity allowed us to draw samples from different socio-cultural regions with varied exposure to technological innovation and across a diverse background of participants to capture the comprehensive acceptance of mHealth.

### Study participants

Twenty-six women (aged 23 to 31) were interviewed; most of them were students, and the majority completed their undergraduate studies in various faculties. They were selected based on their familiarity and exposure to mHealth services using snowball sampling, allowing the researchers to include respondents from diverse backgrounds, considering the fact that mHealth and other digital health services are predominately discovered and used by young and educated consumers [61]. As a network-based convenience sampling, Snowball sampling enabled researchers to draw samples from hard-to-reach areas or unknown parameters [62], such as the current study.

## Study procedures

The primary data was gathered through in-depth interviews (IDIs) using semi-structured interview guidelines based on this research's theoretical framework and objective. In health-care research, semi-structured questionnaires are the most used qualitative data source [63], which enabled researchers to dive deeper into respondents' perceptions and intentions for adoption and use. The questions in IDIs were established considering a broad literature review and focused on participants' overall perceptions and experiences. Before going to the final data collection phase, researchers conducted 3 full-fledged interviews from the sample group to ensure the construct, face and criterion validity. A major outcome was to incorporate rewording of study questions considering sensitivity of the issues being discussed. The planned interview lasted around 25–35 minutes with the participant which was conducted by one researcher (MJ) and a research assistant (SS) recorded the conversation on a smartphone with the consent from the participant and took field notes. Along with field notes, recorded conversations were stored and transcribed by the researchers after listening to them to increase the reliability of data for analysis. Despite participants' varied experiences regarding mHealth and reproductive health, a consistent pattern emerged from 26 interviews. This pattern indicated data saturation and that we had gathered enough data to conclude the data collection process.

## Data analysis

Transcribed interviews were organized with corresponding interview questions. The data underwent analysis using the six phases of the thematic analysis process outlined by Braun & Clarke [64]. The study of transcriptions of the data revealed 11 subthemes. Again, those sub-themes were analyzed and checked against interview data to obtain six themes that emerged as factors of attitude and behavioral intention, namely, (i) belief and perceived outcome, (ii) trust in mHealth, (iii) hesitancy in direct consultation, (iv) exposure to mHealth, (v) self-awareness, and (vi) technological skills. Interpretations were drawn by comparing them with the secondary data collected from journal articles, books, government and non-government reports, and information from websites and newspapers.

## Ethical considerations

The study obtained IRB approval from the Research Review and Ethics Committee of the Department of Public Administration, University of Dhaka (ERC 2019/12), conforming to the Helsinki Declaration guidelines. Before the interview, written informed consent was obtained from each participant voluntarily. Each participant was advised of the study's aims, methods, affiliations, benefits, and hazards at the planned interview session. Confidentiality of respondents' information was maintained throughout the study process. Researchers used unique codes to hide the actual identity of the respondents while gathering information from them. The participants were allowed to terminate the interview at any given moment. Nevertheless, every participant agreed to participate in the interview along with the permission to record the conversion for transcription purpose only.

## Findings

Awareness-raising initiatives of female reproductive functionality and health are a cornerstone for the success of reduction in under-five and maternal mortality, improvement in prenatal and postnatal care, better family planning, and containing deadly sexually transmitted infections like HIV in Bangladesh. According to the findings from the interviews, young, educated, and technologically knowledgeable people were the main customers of mHealth services.

They, as friends and family members, encourage others to adopt mHealth. It was also found that different mobile health services vary in cost. For example, SMS and IVR-based services are more affordable than doctor consultations and diagnostics.

However, mHealth solutions shared a similar design where services were provided through mobile devices or smartphone apps. To avail themselves of services, users ought to register with their phone number and other demographic information regarding the expected service, which includes age, gender, geographic location, and the nature and duration of the problem. Users could choose services at a definite interval, such as daily, weekly, or monthly. There were options for emergency contact in each service, which a dedicated call center of the provider attends. However, respondents considered that services received through SMS and IVR had a limited scope of customization compared to consultative services. But mobile phone operators had the most extensive subscriber base to mHealth in Bangladesh.

In efforts to increase digitalization in healthcare, the government has also invested in Upazila and district-level hospital infrastructures, most prominently *Shastho Batayan*. Other hospitals and diagnostic centers, especially in Dhaka City (such as Popular, Ibn Sina, Square, Lab Aid, and BIRDEM), also provide medical consultations and diagnostics reports through dedicated apps and websites, which are absent outside Dhaka City. However, concerns remained about the availability of expert doctors and qualified medical representatives in the mobile healthcare ecosystem. Although physical consultative services ensure such quality, there is no proper mechanism to ensure quality in the mHealth app, which is sometimes available for free. The issues of data security and the privacy of patients had been significant, according to the findings. Nevertheless, mobile health services provide positive health outcomes regarding awareness, diagnosis, prevention, and control of illness and better family planning among its users. It also saves time and money, as well as visits to healthcare facilities, and safeguards the efficient use of resources, opening the avenues for improved digital health equity, especially for women.

Furthermore, respondents had termed incorporating technology, especially mobile phones or smartphones, in reproductive mobile health services valuable and effective. However, to create a positive attitude and encourage the adoption and use of reproductive mHealth services, the belief of its users towards the technology and perceived impacts was a strong determinant. According to our findings, there have been primarily positive feelings about mHealth services among young women, which they termed "helpful, useful, and user-friendly" services. mHealth has the potential to ensure medical care, comprehensive sexual education, and women's ability to make their own choices and decisions in life, all of which ultimately empower their health decisions. Nevertheless, they believed that accepting mHealth for breaking the societal taboo can be the country's first step towards comprehensive reproductive healthcare.

Most of the participants in the study expressed a moderate to high level of trust in mHealth initiatives, projects, and processes as a dependable source of reproductive healthcare services in Bangladesh. Mobile applications for reproductive health (e.g., period tracker) and vitals monitoring devices (e.g., smartwatch) were trendy among young women. Reproductive information-related mobile applications served as a first base of knowledge on reproductive healthcare among young girls and women, according to a respondent. In Bangladesh, girls are generally shy to share or talk about their menstruation or personal health problems with family members, peers, or even health professionals. However, there are mixed experiences of face-to-face interactions among respondents. An unmarried woman described her experience as positive, while a married woman said that the knowledge from the consultation helped her 'lead a safe conjugal life.' Contrastively, another unmarried young woman described her experience as 'traumatic' due to embarrassment in that setting.

In this context, mHealth was a more viable solution for convenience, given the knowledge and options to choose reproductive advice from anywhere in the world. Even in a conservative society, attitudes toward reproductive mobile health services could be reliable as they protect secrecy and are non-judgmental. In this regard, a participant said,

*Sexual and reproductive health issues are taken to be a taboo subject in our country, and people generally do not talk about this openly. These mobile health services can promote these issues and make the younger generation aware of them.* (IDI 12)

Consequently, mHealth was popularized for its perceived benefits and more accessible patients and diagnosis information access. Although mHealth services are not as widespread as in developed countries, the focus on reproductive healthcare services is narrower. Several sources or factors significantly influenced users' perception and adoption of mHealth services in Bangladesh. According to the findings, the role of peers and social media was the most influential behind adoption. Other sources of influence included web or in-app advertisements, automated short messages from operators or service providers, and social media advertisements. Peers and family members significantly impacted the spread of the advantages of mHealth services and motivated others to adopt and utilize them. Moreover, respondents were generally willing to use or continue using the mHealth service for their reproductive health. However, a few were skeptical about its effectiveness over face-to-face consultations with doctors. Another respondent who used a mobile application-based reproductive health service expressed her satisfaction, like others who regularly use such applications. She said,

*I do and will use several mobile applications and often recommend them to others. It makes us health-conscious and tracks our menstrual cycle, calorie intake, usage, etc. And many other things that usually slip our minds. Having an app to do the heavy lifting is pretty useful.* (IDI 11)

Findings portrayed that self-awareness plays a crucial role in sexual and reproductive health and well-being. It also could be an influential factor as one may not understand the benefits and possibilities of mobile health services. However, a respondent disagreed and attributed an increase in mHealth service adoption to 'anxiety, shyness, and social stigma' rather than self-awareness and anonymity. Others believe self-awareness might help develop a positive attitude towards mHealth and increase adoption and usage through confidence building. A respondent explained her journey with mHealth apps, saying,

*I was facing hormonal changes and missed periods for two consecutive months. Then, I talked to a gynecologist and started following her advice. I started to use apps to keep my period routines and symptoms. When I used to keep track of my period manually, I often mixed up the dates, but with the help of the app, now I get a journal of my period and overall health journey for the last three years.* (IDI 23)

Respondents believed that when people are careful about their health, they use mobile health services more often. Also, it helped their reproductive health decisions. The role of both sexual and reproductive health and technological knowledge was crucial for shaping the attitude regarding mHealth as it allowed the adoption and usage, according to respondents. After tertiary education, maturity and familiarity with the latest technology helped them avoid confrontation with social norms and taboos in their private lives. Interestingly, not only does knowledge of reproductive health and technology usage influence the adoption of reproductive

mHealth services, but those services also have an immense impact on enriching the users' knowledge and attitude. A student summarized the relationship and said,

> *Mobile health services help you to understand your body and health at a primary level. Having adequate sexual and reproductive health knowledge can increase awareness of the importance of seeking and utilizing health services, including reproductive health services. This knowledge can also help them understand the benefits of using mHealth services, such as the convenience and privacy they offer.* (IDI 8)

However, due to Bangladesh's socio-economic, cultural, and technological factors, mHealth may also possess some challenges. It was observed that mHealth services help spread health information and essential medical tips. A significant part of the population in Bangladesh is uneducated. And if they get misinformation or wrong medical services, it would be dangerous. Nonetheless, mHealth can be considered an effective means to boost the status of women's health by utilizing the digital platform the government has laid out in the last decade.

## Discussion

The findings indicated that individuals who utilize mHealth services perceive them as more convenient and secure in comparison to physically visiting healthcare facilities, particularly for minor concerns and inquiries related to their reproductive health. Although the promotion of such services was lagging behind traditional healthcare, the attitude toward reproductive health services in Bangladesh was generally favorable. Thus, the trend was reflected in the increasing adoption and use of mHealth services, as discovered from the participants' experiences. The most influential factors behind their adoption were the helpfulness, usefulness, and effectiveness of mobile health services encouraged by their peers, friends, relatives, and promotions on social media. It has been seen as an essential tool to break societal taboos around women's reproductive health and produces trust for mHealth initiatives.

Furthermore, girls and women in Bangladesh were reserved to share or discuss their menstruation and reproductive health problems due to socio-cultural factors and stigmatization in society. With the economic and social empowerment of women and the rise in educational attainments, there is a tendency to break away from the patriarchal constructs that exist in the current society. They are more eager to learn and maintain good health, including reproductive health. However, reproductive services in Bangladesh were not widespread compared to developed countries, and there is a lack of investment in developing such infrastructures. Yet, based on the theoretical model, it can be assumed that attitude and self-efficacy for intention to use mHealth showed favorable signs while the norms changed in a favorable direction in Bangladesh. Though culture and religion, education and knowledge, and socio-economic status were taking a turn in women's favor, this study showed that beliefs and trusts, age, awareness and experience, and concerns for privacy and data remain more complicated parts of the mHealth adoption.

The findings conformed to a previous study by Choudhury [61], who found that traditional reproductive healthcare services in Bangladesh are mired by various challenges that lower women's access to safe reproductive practices, including social and religious taboos, gender-based discrimination, and disinformation or misinformation associated with women's reproductive functions. Besides, healthcare professionals in Bangladesh viewed the digitalization of the health sector favorably, while doctors and pathologists were prepared to provide digital health services [65]. Thus, mHealth can give opportunities to avail reproductive consultations and services with confidentiality at the ease of their homes. The perceived positive outcome

from mHealth interventions corroborated the findings of Jahangir *et al.* [66], who stated that it could significantly affect the sexual health and well-being of women in Bangladesh. The fact that mHealth was observed to augment self-confidence in selecting family planning [67] and even awareness of the ovulation period led to more and better contraception use [68].

Many mobile apps let women track their menstruation cycle, as regularly used by participants of this study, empowering them to make better reproductive decisions. Other than menstruation and ovulation tracking, women put more trust in mHealth for information seeking on sexual health and reminders for medicine intake. Despite the positive connotation of mobile health services, Ahmed *et al.* argued that digital health innovations in Bangladesh and similar contexts must take socio-demographic attributes and lack of awareness, knowledge, personal comfort, and acceptance [12]. The findings portrayed a similar context regarding mHealth services in Bangladesh, especially for reproductive healthcare, because of the service quality of healthcare facilities and the shyness among girls to talk about sexual health. Moreover, to generalize from participants' experiences, young women are eager to steer clear of those taboos and cultural stigma around reproductive health for their well-being.

Yet, as indicated in previous studies [12,14,15,37,61], sometimes women are not well aware of their own sexual and reproductive health and rights, which not only creates hesitancy to seek traditional reproductive services but also disempowers them to make effective decisions because of a lack of knowledge. Another aspect of the findings showed that peers play a catalytic role in changing the conservative mindsets among potential users of reproductive mHealth services. In the context of rural Bangladesh, Ginsburg *et al.* [69] conducted a randomized controlled trial for clinical attendance for breast cancer symptoms using a mHealth model. According to the results of that study, community health workers with smartphones and patient navigation training were more effective in encouraging women to adhere to advice for clinical attendance compared to the pen-and-pencil method, which indicated that in a resource-poor conservative setting, mHealth could be used as a tool to encourage and promote women's health.

Still, mobile phones can be a critical component to access and strengthen healthcare systems in Bangladesh despite the lack of human resources and infrastructure [24]. According to a study by Ahmed *et al.* [12], only a tiny portion of the population (7.2%, N = 471) used a mobile phone to seek health-related information or services. Most refrained from it due to unfamiliarity, lack of technological competence and literacy, cost of access, lack of awareness, and proximity of physical healthcare facilities. According to other research, that portion of the population shrinks further in the rural context [70]. It was pointed out that despite willingness and readiness to use mHealth, there is inadequate workforce and technical capabilities for people to stimulate the utilization of digital health services [71].

Besides, it was observed that young women searched extensively before consulting a doctor for their reproductive health. Many did not consult a doctor because they had their answers already. So, it implies that by developing a reliable platform for reproductive health services, unnecessary visits to hospitals will be reduced, and the quality of life will be improved, ensuring universal healthcare. This idea aligned with other researchers [72,73], who stated that mHealth is a powerful tool to transform the lives of young people and support their behavioral change. However, in the Bangladeshi context, Islam *et al.* [45] found that the unsatisfactory usability of mHealth applications may pose a barrier to their widespread adoption on a national scale. Moreover, concerning the data security of mHealth apps used by women, Alfawzan *et al.* [74] found that 'many of the most popular' apps have poor data privacy, data sharing, and data security standards, while protocols are not practiced thoroughly. Comparable concerns were raised in our study, but young adult women nonetheless use those apps.

Other than privacy and security challenges, participants in our research had also mentioned that the authenticity of the information, lack of mHealth campaign initiatives, lack of

equipment and technology in hospitals, lack of sex education in schools, and outdated apps and websites were some obstacles that constrain the widespread use of reproductive mHealth services in Bangladesh. Notwithstanding these challenges, the field of mHealth has expanded in Bangladesh with the push for a digital healthcare system as outlined in the NDHS. The analysis of the interview data paints that reproductive mHealth services have tremendous potential to serve all women across the country with minimum access to the internet and smartphones. The patronage and promotion of such services can help to empower women about their reproductive health decisions in a conservative society, even without significant remodeling of the healthcare system.

## Conclusion

The fundamental value of mHealth is contributing to health system goals and outcomes. In doing so, mHealth opens countless possibilities to deliver health services at the fingertip of its users, including a personalized reproductive health service system. Thus, promoting mHealth services at the national level and in local communities could play a pivotal role in its success in improving women's digital health equity in the country. Promotion will be crucial in reforming conservative norms, taboos, and misconceptions about women's health and such endeavors must be initiated by the policy makers. The findings of this study also affirmed and showed an optimistic attitude due to anonymity, reduced cost, easy access, availability of mobile applications, and incorporation of ICT-based services in hospitals. Still, a cultural tension exists in Bangladesh that prevents women from seeking reproductive healthcare independently in a patriarchal structure stigmatizing women's health. These factors contribute to the limited adoption and use of reproductive health services in Bangladesh. Besides, a sustainable userbase for mHealth services is threatened if the cultural barriers regarding technology adoption and normalization of reproductive health are not removed. Hence, promoting reproductive mHealth services through state apparatus could significantly increase adoption and usage. In contrast, a combination of strategic goals to provide reproductive services through the use of mobile phones can further help change healthcare norms in society toward more equitable health outcomes, considering gender.

The NDHS, though, only mentioned mobile health as one of the tools for digitalization of healthcare, there is a substantive need for specific policies for regulations of emerging mHealth, eHealth and telehealth market segments in Bangladesh. Therefore, the findings of this research can be crucial to understanding and designing such policies, projects, and services that would utilize mHealth for reproductive healthcare in the future, as well as drawing its implications in establishing an equitable healthcare system for women. To enhance women's access to sexual health services in a developing nation like Bangladesh, policymakers, healthcare providers, and key stakeholder groups would benefit from understanding public sentiment regarding mHealth and the sociocultural factors that influence its utilization. Notwithstanding potentials benefits, this research suffers significant limitation in terms of women-only samples, low sample size, narrow focus on mHealth users and failure to include perspectives from healthcare providers. While the lack of funding limited the data collection in seven places and among young and educated participants, future research can be undertaken considering these limitations. Also, researchers suggest that more research is needed to determine the usage patterns of reproductive mHealth services and their effectiveness especially in quantitative manner that would identify implementation challenges in terms of customization and personalization in reproductive healthcare in a developing country like Bangladesh. Nonetheless, awareness and promotion would play a key role in encouraging the adoption of these

services in rural communities and marginalized sections of society and educating them on safe and healthy lives instead of taboos and misconceptions.

## Supporting information

**S1 File. Transcription of responses from the IDIs.**
(DOCX)

## Acknowledgments

The authors would like to thank the research assistant, Sharmin Sultana (SS), for her untiring support in the data collection process.

## Author Contributions

**Conceptualization:** M. Jonayed, Maruf Hasan Rumi.

**Data curation:** M. Jonayed.

**Formal analysis:** M. Jonayed.

**Methodology:** M. Jonayed.

**Project administration:** M. Jonayed.

**Resources:** M. Jonayed, Maruf Hasan Rumi.

**Validation:** M. Jonayed.

**Writing – original draft:** M. Jonayed.

**Writing – review & editing:** M. Jonayed, Maruf Hasan Rumi.

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
