## [Decision Letter · Decision Letter 0]

5 Aug 2024

PDIG-D-24-00267

Towards women’s digital health equity: A qualitative inquiry into attitude and adoption of reproductive mHealth services in Bangladesh

PLOS Digital Health

Dear Dr. Jonayed,

Thank you for submitting your manuscript to PLOS Digital Health. After careful consideration, we feel that it has merit but does not fully meet PLOS Digital Health's publication criteria as it currently stands. Therefore, we invite you to submit a revised version of the manuscript that addresses the points raised during the review process.

Please submit your revised manuscript within 60 days Oct 04 2024 11:59PM. If you will need more time than this to complete your revisions, please reply to this message or contact the journal office at digitalhealth@plos.org. Please include the following items when submitting your revised manuscript:

We look forward to receiving your revised manuscript.

Kind regards,

Haleh Ayatollahi

Section Editor

PLOS Digital Health

Journal Requirements:

Additional Editor Comments (if provided):

Reviewers' comments:

Reviewer's Responses to Questions

**Comments to the Author**

1. Does this manuscript meet PLOS Digital Health’s publication criteria? Is the manuscript technically sound, and do the data support the conclusions? The manuscript must describe methodologically and ethically rigorous research with conclusions that are appropriately drawn based on the data presented.

Reviewer #1: Yes

Reviewer #2: Yes

Reviewer #3: Partly

2. Has the statistical analysis been performed appropriately and rigorously?

Reviewer #1: Yes

Reviewer #2: Yes

Reviewer #3: N/A

3. Have the authors made all data underlying the findings in their manuscript fully available (please refer to the Data Availability Statement at the start of the manuscript PDF file)?

Reviewer #1: Yes

Reviewer #2: Yes

Reviewer #3: No

4. Is the manuscript presented in an intelligible fashion and written in standard English?

Reviewer #1: Yes

Reviewer #2: Yes

Reviewer #3: Yes

5. Review Comments to the Author

Reviewer #1: • The manuscript provides invaluable insights into women's attitudes towards reproductive health services in Bangladesh, a topic of urgent concern. It fills a crucial gap in the literature and underscores the potential impact of this research, offering hope for significant improvements in women's reproductive health.

• The Integrative Model of Planned Behavior (IMPB) is not just well-justified, but its application is also reassuringly appropriate, enhancing the credibility of the research.

• The findings highlight and illuminate essential themes such as the convenience and security of mHealth services, trust issues, and socio-cultural barriers.

• The sample size is small and limited to university students, which may not fully represent the diverse experiences of women across Bangladesh. Future research should include a more varied demographic.

• More detail on ensuring the reliability and validity of the findings would strengthen the study.

• Specific, actionable recommendations for policymakers and healthcare providers are beneficial and crucial for enhancing the study's practical utility and improving women's reproductive health in Bangladesh.

• The manuscript could benefit from a more critical discussion of the findings and their implications.

Reviewer #2: The analysis was done as required and the manuscript is presented in an acceptable fashion. what can be improved is adding an explanation on the ethical considerations beyond obtaining consent, such as ensuring confidentiality, handling sensitive information, and addressing any potential risks or conflicts of interest, which are crucial aspects in qualitative research involving human subjects.

Reviewer #3: Despite being a significant topic, the major concern associated with the submitted manuscript is the lack of proper sample size selection. Only 26 cases are not sufficient to provide any sort of interpretation. Additionally, the study seems to be more on review based study than the original research, so it could be applied as a review paper and not an original research.

6. PLOS authors have the option to publish the peer review history of their article (what does this mean?). If published, this will include your full peer review and any attached files.

**Do you want your identity to be public for this peer review?** For information about this choice, including consent withdrawal, please see our Privacy Policy.

Reviewer #1: No

Reviewer #2: No

Reviewer #3: No

---

## [Decision Letter · Decision Letter 1]

10 Sep 2024

Towards women’s digital health equity: A qualitative inquiry into attitude and adoption of reproductive mHealth services in Bangladesh

PDIG-D-24-00267R1

Dear Mr. Jonayed,

We are pleased to inform you that your manuscript 'Towards women’s digital health equity: A qualitative inquiry into attitude and adoption of reproductive mHealth services in Bangladesh' has been provisionally accepted for publication in PLOS Digital Health.

Best regards,

Haleh Ayatollahi

Section Editor

PLOS Digital Health

Reviewer Comments (if any, and for reference):

Reviewer's Responses to Questions

**Comments to the Author**

1. If the authors have adequately addressed your comments raised in a previous round of review and you feel that this manuscript is now acceptable for publication, you may indicate that here to bypass the “Comments to the Author” section, enter your conflict of interest statement in the “Confidential to Editor” section, and submit your "Accept" recommendation.

Reviewer #1: All comments have been addressed

Reviewer #2: All comments have been addressed

Reviewer #3: All comments have been addressed

2. Does this manuscript meet PLOS Digital Health’s publication criteria? Is the manuscript technically sound, and do the data support the conclusions? The manuscript must describe methodologically and ethically rigorous research with conclusions that are appropriately drawn based on the data presented.

Reviewer #1: Yes

Reviewer #2: Yes

Reviewer #3: Yes

3. Has the statistical analysis been performed appropriately and rigorously?

Reviewer #1: Yes

Reviewer #2: Yes

Reviewer #3: N/A

4. Have the authors made all data underlying the findings in their manuscript fully available (please refer to the Data Availability Statement at the start of the manuscript PDF file)?

Reviewer #1: Yes

Reviewer #2: Yes

Reviewer #3: No

5. Is the manuscript presented in an intelligible fashion and written in standard English?

Reviewer #1: Yes

Reviewer #2: Yes

Reviewer #3: Yes

6. Review Comments to the Author

Reviewer #1: (No Response)

Reviewer #2: N/A

Reviewer #3: All comments of the reviewers are addressed.

7. PLOS authors have the option to publish the peer review history of their article (what does this mean?). If published, this will include your full peer review and any attached files.

**Do you want your identity to be public for this peer review?** For information about this choice, including consent withdrawal, please see our Privacy Policy.

Reviewer #1: **Yes: **Dr K Madan Gopal

Reviewer #2: No

Reviewer #3: No
